# In Vitro and In Silico Wound-Healing Activity of Two Cationic Peptides Derived from Cecropin D in *Galleria mellonella*

**DOI:** 10.3390/antibiotics14070651

**Published:** 2025-06-27

**Authors:** Sandra Patricia Rivera-Sanchez, Iván Darío Ocampo-Ibáñez, Maria Camila Moncaleano, Yamil Liscano, Liliana Janeth Flórez Elvira, Yesid Armando Aristizabal Salazar, Luis Martínez-Martínez, Jose Oñate-Garzon

**Affiliations:** 1Research Group of Microbiology, Industry and Environment, Faculty of Basic Sciences, Universidad Santiago of Cali, Cali 760035, Colombia; ivan.ocampo00@usc.edu.co; 2Transnational Research Group on Infectious Diseases, PhD School of Biomedicine, University of Córdoba, 14071 Córdoba, Spain; 3Research Group of Chemistry and Biotechnology, Faculty of Basic Sciences, Universidad Santiago of Cali, Cali 760035, Colombia; maria.moncaleano00@usc.edu.co (M.C.M.); yesid.aristizabal00@usc.edu.co (Y.A.A.S.); 4Research Group of Comprehensive Health (GISI), Department Faculty of Health, Universidad Santiago de Cali, Cali 760035, Colombia; yamil.liscano00@usc.edu.co; 5Health Faculty, Universidad del Valle, Cali 760042, Colombia; liliana.florez@univalle.edu.co; 6Microbiology Unit, Reina Sofía University Hospital, 14008 Córdoba, Spain; luis.martinez.martinez.sspa@juntadeandalucia.es; 7Maimonides Institute for Biomedical Research of Córdoba, 14008 Córdoba, Spain; 8Department of Agricultural Chemistry, Soil Sciencies and Microbiology, University of Córdoba, 14071 Córdoba, Spain; 9Centro de Investigación Biomédica en Red de Enfermedades Infecciosas (CIBERINFEC), Instituto de Salud Carlos III, 28040 Madrid, Spain

**Keywords:** cationic peptides, cecropin D, in vitro, molecular docking, tissue regeneration, wound healing

## Abstract

Background: Chronic wounds pose a significant public health challenge due to high treatment costs and the limited efficacy of current therapies. This study aims to evaluate the in vitro wound-healing activity and in silico interactions of two antimicrobial cationic peptides, derived from *Galleria mellonella* cecropin D, whose receptors are involved in tissue healing. Methods: Two peptides were tested: a long peptide (∆M2, 39 amino acids) and a short peptide (CAMP-CecD, 18 amino acids). Their cytotoxicity, as well as their effects on fibroblast proliferation and migration, were assessed using Detroit 551 cells. In parallel, molecular docking studies were conducted with AutoDock Vina to predict the binding affinities of these peptides to the key receptors involved in wound healing: the epidermal growth factor receptor (EGFR), the transforming growth factor beta receptor (TGFRβ2), and the vascular endothelial growth factor receptor (VEGFR). Results: In vitro assays showed that the short peptide exhibited lower cytotoxicity and significantly enhanced cell proliferation and migration, leading to a greater percentage of gap closure compared to the long peptide. A docking analysis revealed binding affinities of −6.7, −7.2, and −5.6 kcal/mol for VEGFR, EGFR, and TGFRβ2, respectively, with the RMSD values below 2 Å, indicating stable binding interactions. Conclusions: These findings suggest that the structure and cationic charge of the short peptide facilitate robust interactions with growth factor receptors, enhancing re-epithelialization and tissue regeneration. Consequently, this peptide is a promising candidate ligand for the treatment of chronic wounds and associated infections.

## 1. Introduction

Chronic wounds, including but not limited to diabetic foot ulcers, pressure ulcers, and other lower-limb ulcers, represent a significant public health concern, primarily due to the substantial financial burden they place on healthcare systems [1,2,3]. The process of wound healing is an inherently intricate process comprising four phases (hemostasis, inflammation, proliferation, and remodeling) with multiple cellular and biochemical mechanisms [4]. The process of hemostasis involves the formation of a fibrin clot, which functions to prevent further bleeding. In the inflammatory phase, the release of cytokines and growth factors (e.g., EGF, PDGF, FGF, and TGF-β) by neutrophils and macrophages is crucial for the clearance of pathogens and debris [5,6,7]. Subsequently, during the proliferative phase (days 2–21), the generation of new blood vessels and collagen fibers occurs. Ultimately, in the remodeling phase (days 21 to one year), these fibers undergo reorganization and replacement, thereby restoring tissue strength [8,9].

Growth factor receptors, which include EGFR, TGFRβ2, and VEGFR, have been identified as the primary mediators of these events [10,11]. For instance, the TGFβ–TGFRβ2 complex has been shown to initiate signaling pathways involving SMAD3/SMAD2 and modulate cellular proliferation depending on the environment and the presence of amino acids like arginine [12,13]. In a similar manner, EGF has been observed to bind to EGFR, thereby promoting skin regeneration [14]. By contrast, the VEGF–VEGFR pathway has been shown to activate angiogenesis through the action of serine/arginine-rich domain proteins [15,16]. Exogenous substances, such as cationic antimicrobial peptides (CAMPs), have the potential to promote intracellular signaling due to their demonstrated capacity for healing effects [17,18,19,20]. The effects of LL-37 on cell proliferation, migration, and tissue regeneration have been demonstrated in several studies [21,22]. Furthermore, these peptides have been shown to modulate the pathways associated with both angiogenesis and the regulation of inflammation [23,24]. Previous studies have reported that peptides, such as P-LL3, promote wound healing in mice by activating re-epithelialization and vascularization pathways [25]. Similarly, the peptide, PMPP, administered topically to mouse tissue, significantly promoted the wound epithelialization process compared to the control group, and completed wound healing on day 14 [26]. Cecropin D, derived from *Galleria mellonella*, is of particular interest given its non-toxic activity against multidrug-resistant pathogens and its potential immunomodulatory effect. These properties make it attractive for wound-healing applications [27,28,29]. It was used as a template sequence for the design and synthesis of two synthetic cationic peptides (CAMP) derived from cepropin D, which were evaluated in this research. The first, called ∆M2, is a long peptide (LongPep). The second peptide, CAMP-CecD, is an analogue of cecropin D derived from the N-terminal region of ∆M2, a short peptide (ShortPep). Both were considered for the study of healing in fibroblasts due to the presence of arginines in their primary structure, an amino acid involved in the activation of EGFR receptors, which could accelerate re-epithelialization and wound closure during healing [18]. Despite its promising healing activity, its widespread use in therapeutic interventions is limited by concerns regarding toxicity, stability, and cost [30].

Molecular docking is a systematic bioinformatic approach that can be used to analyze the interaction between these peptides and growth factor receptors (EGFR, TGFRβ2, and VEGFR). It can be used to predict the binding modes and affinities between ligands (peptides, in this case) and their receptors [31]. This method uses algorithms to calculate atomic-level interaction forces. This provides insight into the possible signaling pathways and activation mechanisms involved in tissue repair.

Therefore, this study investigated the in vitro wound-healing activity of two cationic peptides derived from *Galleria mellonella* cecropin D on skin fibroblasts, using proliferation, cytotoxicity, and cell migration assays. Additionally, to further explore their potential mechanisms of action at the level of growth factor receptors (EGFR, TGFRβ2, and VEGFR), an in silico molecular docking analysis was performed to predict the interactions between these peptides and the aforementioned receptors. This comprehensive approach integrates biological experimentation with computational modeling to identify novel therapeutic candidates for tissue regeneration and the treatment of chronic wounds.

## 2. Results and Discussion

### 2.1. Design and Synthesis of Antimicrobial Peptides

This study evaluated two synthetic cationic antimicrobial peptides (CAMPs). The first, designated ∆M2, is derived from cecropin D. It consists of 39 amino acid residues, carries a net positive charge of +9, and has a hydrophobicity index of 0.178 [19,32].

The second peptide, CAMP-CecD, is a cecropin D analog derived from the N-terminal region of ∆M2. It consists of the first 18 amino acid residues of ∆M2, retains a net positive charge of +9, and has a hydrophobicity index of −0.002. An analysis of its net charge and helical wheel projection revealed a distinctly amphipathic configuration. The hydrophilic region comprises 10 residues (R1, R6, R8, R9, R13, R15, K5, K12, K16, and N2), while the hydrophobic core includes 8 residues (A10, A17, F3, F4, I7, I14, I18, and G11) [33]. These physicochemical and structural properties regulate the antimicrobial activity, mechanism of action, and toxicity profile of these peptides [34].

Both peptides exhibit α-helical structures and, due to their net positive charge of +9, are capable of forming electrostatic interactions with negatively charged phospholipids, such as cardiolipins, found in the mitochondrial and bacterial membranes [34,35]. Although the longer peptide is more hydrophobic than the shorter one, its hydrophobicity remains below 50%, suggesting low hemolytic activity in mammalian cells without compromising its antimicrobial properties [36]. In addition, the sequence and amino acid composition contribute to peptide toxicity. Highly α-helical peptides containing lysine and/or leucine, alanine, and glycine in sequences of up to 21 residues tend to exhibit cytotoxicity in mammalian cells [34,37]. In this context, both peptides in the present study contain arginine and glycine residues, but the longer peptide includes a greater number of lysines. Specifically, the longer peptide contains six arginines, five lysines, and three glycines, whereas the shorter peptide includes six arginines, three lysines, and one glycine. Notably, the presence of arginine is of particular interest, as this residue has been shown to activate EGFR receptors, thereby promoting re-epithelialization and wound closure during the healing process [18].

### 2.2. Cytotoxicity Assays

The cytotoxicity assay was evaluated in the Detroit 551 human skin fibroblasts after 72 h of incubation by measuring the percentage of cell viability in the presence of the CAMPs (short and long) at serial concentrations ranging from 1 µg/mL to 1024 µg/mL, compared to the untreated control cells. The results, presented in Figure 1, correspond to the averages of two independent assays performed in triplicate. The values were normalized with respect to the number of live cells in the control wells, considered to have 100% viability.

According to the results in Figure 1, the long peptide showed a greater cytotoxic effect than the short peptide at high concentrations, while at low concentrations (between 1 and 4 µg/mL), no cytotoxicity was observed for either peptide, with viability percentages even higher than those observed for the cells not exposed to the peptides (viability control), as is the case with the long peptide, which, at a concentration of 4 µg/mL, interestingly promoted an increase in cell viability above 100%. On the other hand, the cytotoxic effect was evident for the long peptide at a concentration of 64 µg/mL, with an average viability percentage of 70.8%, while for the short peptide, the cytotoxic effect began to be evident at a concentration of 256 µg/mL, with an average viability percentage of 62.8 µg/mL.

Continuing with the analysis of the results in Figure 1, when comparing the viable cells treated with the long peptide to the untreated cells, statistically significant differences were found up to a concentration of 16 µg/mL, with a *p*-value < 0.001, followed by a concentration of 4 µg/mL with statistically significant differences, with a *p*-value ≤ 0.01. Statistically significant differences were also found, with a *p*-value < 0.001 for the concentration of 1024 µg/mL and with a *p*-value < 0.01 for the concentration of 256 µg/mL when comparing the untreated cells with the viable cells treated with the short peptide.

An increase in cell viability above 100% was obtained at the low concentration (4 µg/mL) of the long peptide together with the MTT compound, suggesting that, in conjunction with the MTT compound, mitochondrial enzyme metabolism was stimulated. In this process, the enzyme succinate dehydrogenase reduces MTT to insoluble formazan, indicating a positive adaptive response or hormesis [38,39]. It should be noted that hormesis favors proliferation and functional connection between mitochondria, promoting effective mitochondrial biogenesis, since exposure to mild stimuli activates biochemical pathways that improve the remodeling and efficiency of the mitochondrial network [39,40].

However, cytotoxicity becomes evident at high concentrations. For the long peptide, it was found at a concentration of 64 µg/mL, while for the short peptide, cytotoxicity was evident at a concentration of 256 µg/mL. According to Martens et al. (2019) and Heil et al. (2017) [41,42], a treatment that reduces viability below 80% is considered cytotoxic, suggesting that both peptides induce cell death at high concentrations [43]. This is related to the fact that, at concentrations of 64 µg/mL for the long peptide and 256 µg/mL for the short peptide, damage to the cell membrane occurs, preventing adequate retention of the formazan generated in the assay [43].

The finding of cytotoxicity at high concentrations is particularly relevant in the context of wound healing, a metabolically demanding process in which mitochondria play a crucial role in ATP generation and in providing metabolic precursors for cell proliferation and migration [44]. It is important to note that proliferating cells can resort to alternative energy production mechanisms, such as aerobic glycolysis, which complement mitochondrial function in the biosynthesis of citrate, a key intermediate in the tricarboxylic acid cycle and a regulator of cell growth [40].

Both long and short peptides are cytotoxic at concentrations of 64 and 256 µg/mL, respectively. This may be due to their positive charge (+9). The peptides mainly act on membranes, altering their integrity. This leads to adverse cellular processes, ultimately resulting in cell death [34,35]. Compared to the short peptide, the long peptide has a higher hydrophobicity index, a structural characteristic closely linked to the cytotoxic activity of the CAMPs [36]. The degree of partitioning of the peptide within the hydrophobic core of the membrane is governed by hydrophobicity; therefore, the more hydrophobic the peptide, the more likely it is to destabilize the membrane [34].

According to Mojsoska and Jenssen (2015) [34] certain amino acids, such as alanine, leucine, arginine, and lysine, have a high propensity to adopt α-helical conformations. This is why they have been included in the design of the novel CAMPs. Peptides with around 80% helical content exhibit remarkable antimicrobial potential. Similarly, the presence of lysine and/or leucine, alanine, and glycine residues in sequences of up to 21 residues has been observed to increase the α-helical conformation, which correlates directly with toxicity to mammalian cells [34]. This could explain why the long peptide (39 amino acid residues) is more cytotoxic than the short peptide (18 residues). The long peptide contains six arginines, two additional glycines, and two additional lysine residues. By contrast, the short peptide has only three lysine residues despite being formed from the first 18 amino acids of the long peptide.

On the other hand, hydrophobicity is a key structural feature that determines an antimicrobial peptide’s overall activity [34,45,46]. Higher hydrophobicity leads to lower antibacterial specificity and greater toxicity towards mammalian cells [34]. In this study, the short peptide has a hydrophobicity value of −0.002, while the long peptide exhibits a value of 0.178 (equivalent to 46.3%). Since this value does not exceed 50%, it is assumed that the long peptide does not exhibit high hemolytic activity against mammalian cells, and maintains its antimicrobial efficacy [34,36,47]. This is consistent with the observation that the long peptide is cytotoxic to fibroblast mitochondria at concentrations above 64 µg/mL, which highlights the importance of its physicochemical properties. These same characteristics precisely explain the antimicrobial potential against multidrug-resistant Gram-negative bacteria, such as *Pseudomonas aeruginosa* and *Klebsiella pneumoniae*, as described in the previous studies [32,33].

### 2.3. Proliferation Assays

In the proliferation assay, the effects of the CAMPs on the Detroit 551 fibroblast cells were evaluated. Experiments were conducted at concentrations of 31.2 µg/mL, 125 µg/mL, and 500 µg/mL. Under these conditions, the short peptide exhibited no cytotoxicity or low cytotoxicity at 31.2 µg/mL and 125 µg/mL, respectively, at 72 h. By contrast, both peptides showed cytotoxic effects at 500 µg/mL after 72 h. The fluorescence intensity measured using the alamarBlue™ reagent was higher in the cells treated with the short peptide compared to the control group (untreated cells) at 24, 48, and 72 h, except at the 500 µg/mL concentration at 72 h (see Figure 2A–C). By contrast, the treatment with the long peptide led to a decrease in fluorescence intensity relative to the control, indicating a negative impact on cell proliferation.

A statistical analysis of Figure 2A shows that, at the 500 µg/mL concentration, the short peptide produced higher mean fluorescence intensity values compared to the untreated control group, suggesting that the short peptide promotes proliferation in the Detroit 551 fibroblast-like skin cells. This difference was statistically significant at 24 h, with a *p*-value < 0.001. Conversely, the long peptide showed statistically significant reductions in fluorescence intensity from 24 to 72 h, with mean values consistently lower than those of the untreated cells (*p* < 0.001).

In Figure 2B (125 µg /mL) and Figure 2C (31.2 µg /mL), at all three time points (24–72 h), the short peptide showed higher average cell proliferation, with fluorescence intensity values higher than those obtained for the untreated cells, which were statistically significant, with a *p*-value < 0.001. However, the long peptide showed lower average cell proliferation with fluorescence intensity values lower than those obtained for the untreated cells at the three time points, measured with a *p*-value < 0.001, this difference in averages being statistically significant.

The short peptide promotes the proliferation of the Detroit 551 cells when compared to the growth of cells in the absence of the peptide treatment. Similarly, when comparing the long and short peptides at the three time points (24, 48, and 72 h) and the three concentrations ((A) 500 µg /mL, (B) 125 µg /mL, and (C) 31.2 µg /mL), a higher fluorescence intensity is observed for the short peptide compared to the long peptide when compared to the untreated cells.

The increase in fluorescence intensity is due to the conversion of the alamarBlue™ reagent—chemically known as resazurin (non-fluorescent)—into resorufin (fluorescent), mediated by various redox enzymes that utilize electron donors, such as FADH_2_ or NADPH, which are involved in mitochondrial oxidative phosphorylation [48]. The resulting fluorescence signal is detected at excitation wavelengths of 530–560 nm and an emission wavelength of 590 nm [1], indicating that fibroblast mitochondria are metabolically active. This mechanism, whereby increased mitochondrial metabolic activity leads to greater fluorescence, serves as an indirect indicator of cell viability and proliferation. Accordingly, the results suggest that the short antimicrobial peptide stimulates the proliferation of the Detroit 551 fibroblasts, while the long peptide negatively impacts this process at equivalent concentrations. The higher fluorescence intensity observed in the short peptide treatment may be associated with enhanced mitochondrial activity and increased cellular processes, such as collagen deposition, which is critical for the remodeling phase of wound healing [42].

Furthermore, these findings confirm the proliferative effects of the cationic CAMPs. At high concentrations, the short peptide induces greater proliferation than that observed in the untreated cells, suggesting it may function as an epidermal growth factor (EGF) mimetic [18]. This potential interaction with the epidermal growth factor receptor (EGFR) could activate downstream signaling cascades, enhancing the activity of the key transcription factors, such as c-Myc, c-Fos, and c-Jun, which are involved in regulating the gene expression required for cell proliferation. By contrast, the long peptide may exhibit lower binding affinity to EGFR, thereby limiting the initiation of this signaling cascade. The interaction between EGFR and both peptides will be further explored through in silico analysis later in this study, consistent with the findings of Esquirol Caussa and Herrero Vila (2016) [14], who demonstrated that EGF promotes cell growth, proliferation, differentiation, and survival through its specific receptor (EGFR).

### 2.4. Cell Migration Assay

#### Wound-Healing Potential of the Antimicrobial Peptides

In this study, the effects of both CAMPs (short and long peptides) on cell migration in the Detroit 551 fibroblasts were evaluated. Figure 3 summarizes the progression of wound closure percentage at three concentrations (4 µg/mL, 16 µg/mL, and 64 µg/mL) over different time intervals: 0–6 h, 0–16 h, and 0–24 h.

Statistical analysis presented in Figure 3A, at a concentration of 4 µg/mL, shows that during the 0–6 h interval, the wound closure percentage was approximately 3% for the long peptide and 13% for the short peptide, with no statistically significant differences compared to the untreated control. At 16 h, the long peptide achieved a wound closure percentage of approximately 29%, and the short peptide approximately 38%; however, neither value was statistically different from the control. In the 0–24 h period, the long peptide produced a closure rate of 30%, with no significant difference compared to the control; whereas, the short peptide reached 55% wound closure, showing statistically significant differences compared to the untreated cells (*p* < 0.01).

In Figure 3B, at a concentration of 16 µg/mL, the short peptide promoted wound closure of 27% at 6 h, 49% at 16 h, and approximately 55% at 24 h. These values were statistically significant when compared to the untreated cells, with a *p*-value < 0.05 at 6 h and *p* < 0.01 at 16 and 24 h. By contrast, the long peptide produced wound closure percentages of 15% at 6 h, 30% at 16 h, and approximately 37% at 24 h, with no statistically significant differences from the control at any time point.

In Figure 3C, at a concentration of 64 µg/mL, the short peptide achieved a wound closure of 38% at 6 h, approximately 51% at 16 h, and 78.5% at 24 h. Statistically significant differences were observed at all three time points compared to the untreated control, with *p* < 0.01 at 6 and 16 h, and *p* < 0.001 at 24 h. For the long peptide, significant differences were only observed at 24 h, with a wound closure percentage of approximately 69.5% and a *p*-value < 0.001.

These findings suggest that, after a few hours, fibroblasts migrate toward the plastic insert of the CytoSelect™ system, orienting themselves toward the gap to initiate cell migration. This process occurs in response to various chemotactic stimuli, such as collagen fragments, fibronectin, eicosanoids, elastin, and thrombin, as well as growth factors (PDGF, TGF-β, TNF-α, FGF, and EGF), which stimulate the division, activity, and/or differentiation of fibroblasts [49,50].

It is important to note that initial wound healing is characterized by massive collagen synthesis. During this phase, dermal fibroblasts produce procollagen, which is transformed into collagen, an essential protein of the extracellular matrix (ECM) responsible for maintaining dermal structure. Subsequently, in the remodeling phase, this collagen is rearranged and replaced by a more organized collagen, better prepared to resist mechanical stress [42].

It should be noted that the physicochemical properties of peptides—such as net positive charge, amino acid sequence length, and hydrophobicity—not only determine their antimicrobial activity but influence their immunomodulatory potential. These characteristics can elicit pro-inflammatory, anti-inflammatory, or general modulatory responses, and are best understood as contributors to immune homeostasis by balancing inflammatory processes [22,28,51]. For instance, the cationic peptide LL-37 has demonstrated efficacy in the treatment of venous ulcers, highlighting the relevance of these peptides in tissue repair [23,24]. Moreover, CAMPs have been shown to promote cell proliferation and migration, although their role in angiogenesis appears to be minimal [52].

The LL-37–mediated enhancement of HaCaT keratinocyte migration involves signaling mechanisms distinct from traditional EGFR transactivation, contributing to its effectiveness in wound healing [52]. During the proliferative phase of healing, re-epithelialization of the wound occurs—often reaching 95% closure or more—along with the formation and expansion of granulation tissue. In this phase, fibroblasts play a key role by secreting proteases that degrade the existing extracellular matrix (ECM), and by depositing ECM components, such as collagen, hyaluronic acid, fibronectin, and proteoglycans, which collectively form the scaffold necessary for tissue regeneration [40].

On the other hand, the ShortPep exhibited significantly greater healing activity in vitro (Appendix A), although both peptides showed comparable wound closure percentages at 16 µg/mL after 24 h of incubation. It is plausible that both peptides act through similar pathways but yield different functional outcomes across most assays due to variations in their mechanisms of action, efficacy, and toxicity profiles. The ShortPep, with a net charge of +9 and a pronounced hydrophobic moment (μH = 0.723), likely exerts its superior activity primarily through strong and stable interactions with growth factor receptors, such as EGFR [18]. This, combined with its lower cytotoxicity, results in the enhanced stimulation of cell proliferation and migration—key processes in tissue repair [18].

By contrast, the LongPep, which shares the same net charge (+9) but has a longer sequence and a substantially lower hydrophobic moment (μH = 0.296), appears to rely less on high-affinity receptor binding (as shown later). Despite its lower μH, its markedly higher aliphatic index (95.38 vs. 76.11) and extended peptide length may promote alternative interactions; however, these characteristics also appear to contribute to increased cytotoxicity, thereby limiting its regenerative potential. While the cationic charge of +9 is crucial for the initial interaction with membranes or receptors [34], ultimate efficacy seems to depend on the peptide’s specific affinity for growth factor receptors and its cytotoxicity profile—parameters in which the ShortPep demonstrates a clear advantage.

Differences in physicochemical properties, such as hydrophobic moment and aliphatic index, modulate both receptor interaction and cytotoxic effects. Rather than compensating for its lower receptor affinity and higher toxicity, the LongPep’s greater size and aliphatic index may, in fact, exacerbate these limitations. As a result, despite occasional overlap in functional outcomes, the ShortPep consistently emerges as the more promising candidate across the set of experimental trials.

A key hypothesis in this study is that cationic peptides, by acting as immunomodulators of the innate immune system, could stimulate macrophages in the proliferative phase to secrete growth factors, such as EGF, TGF-β, and VEGF, to promote healing. In particular, their ability to interact with the key receptors, such as EGFR, TGFRβ2, and VEGFR, suggests that these peptides may enhance the cellular signaling necessary for the migration and proliferation of fibroblasts and keratinocytes [34].

In silico docking studies support this hypothesis by demonstrating that both peptides have a high affinity for these receptors, with stronger interactions observed for the short peptide. Similar to the in vitro results, statistically significant differences were observed in the percentage of gap closure for each peptide compared to the untreated cells: 78.5% for the short peptide and 69.5% for the long peptide at a concentration of 64 µg/mL, with *p*-values of <0.001 for both peptides at 24 h. These results are encouraging because the presence of arginines and lysines in their amino acid sequences could favor the activation of growth factors and their interaction with EGFR, a key receptor in cell proliferation [18]. Moreover, VEGF activation plays a crucial role in angiogenesis during re-epithelialization by promoting the proliferation and migration of endothelial cells to the affected area [40]. Our in silico findings confirm that both peptides interact favorably with EGFR, TGFBR2, and VEGFR. The short peptide shows greater affinity for these receptors. These results suggest that combining both peptides could optimize their mechanism of action and enhance the healing process.

In general, cationic peptides not only exhibit wound-healing potential but possess antimicrobial activity, making them particularly attractive for the treatment of infected chronic ulcers, as they could simultaneously promote tissue regeneration and infection control [13,32,33].

### 2.5. Molecular Docking Results

To characterize the interactions of the peptides with the receptors involved in wound healing (VEGFR, EGFR, TGFRβ2), a molecular docking study was conducted using AutoDock Vina [28]. The following well-characterized compounds were used as controls: axitinib (VEGFR), osimertinib (EGFR), and galunisertib (TGFRβ2). Table 1 summarizes the average binding affinities (kcal/mol ± standard deviation) and the average RMSD values (Å) for the best docked poses (all with RMSD < 2 Å).

The short peptide (ShortPep) exhibited a higher affinity than the long peptide (LongPep) for all three receptors. Additionally, both peptides had RMSD values below 2 Å, indicating stable docking and good reproducibility in repeated docking runs. These findings are consistent with in vitro data on proliferation and cell migration, in which the ShortPep induced a stronger wound-healing effect, which is potentially linked to its greater capacity to bind to the epidermal growth factor receptor (EGFR), transforming the growth factor-beta receptor 2 (TGFRβ2) and the vascular endothelial growth factor receptor (VEGFR).

Figure 4 shows the EGFR–ShortPep and EGFR–LongPep complexes, alluvial diagrams (A and B), and an energy decomposition analysis (C and D). Yellow indicates electrostatic interactions, blue denotes hydrogen bonds, and red represents hydrophobic interactions. In the bar charts, orange corresponds to receptor residues and green corresponds to ligand residues. In the EGFR–ShortPep (A) diagram, numerous electrostatic interactions were observed between the arginine residues of the peptide and the aspartic acid residues of the receptor. There were also hydrogen bonds and hydrophobic interactions with phenylalanine, isoleucine, and alanine residues. The energy analysis (Figure 4C) showed that Pro772, Gln820/849, Asp830, His850, and Arg962 in the receptor contribute to stability. In the ligand, Phe3/4, Ile7/14/18, Ala17, and Arg1/6/9 stand out, with Arg9 contributing most to binding (Appendix A). By contrast, in the EGFR–LongPep (B) structure, the contribution of arginine residues is reduced in favor of residues such as Ala, Gly, Ser, and Lys. This weakens the electrostatic interactions. The decomposition diagram (D) revealed that His850 promoted affinity; however, Lys852 and other residues had an unfavorable effect, reducing the total binding energy via intramolecular interactions (Appendix A).

Figure 5 shows the TGFRB2–ShortPep and TGFRB2–LongPep complexes using the same color conventions as the alluvial diagrams (A and B) and energy charts (C and D). In the TGFRB2–ShortPep complex (A), hydrogen bonds are prominent, and electrostatic and hydrophobic interactions are maintained throughout the simulation. The energy analysis (C) indicated that Lys361 in the receptor and Arg1 in the peptide may hinder binding (Appendix A). In TGFRB2–LongPep (B), meanwhile, electrostatic interactions between Arg and Asp in the receptor are more notable, as are hydrophobic interactions involving phenylalanine, proline, and valine (Appendix A). However, as shown in (D), Ala1 ultimately decreased affinity, indicating that its intramolecular participation lowers the complex’s global stability.

Figure 6 focuses on the VEGFR–ShortPep and VEGFR–LongPep complexes, following the same symbolic conventions for the alluvial diagrams (A, B) and the energy graphs (C, D). In VEGFR–ShortPep (A), the contribution of Arg is striking, as it formed electrostatic and hydrogen bonds with Asp and Gln in the receptor; Phe and Ile residues strengthened hydrophobic interactions (Appendix A). The energy analysis (C) showed that Arg929 and Asp1058 in the receptor destabilized the complex, as did Lys5 in the peptide, which interacts intramolecularly and interferes with optimal receptor binding. Finally, in VEGFR–LongPep (B), hydrogen bonds dominated, with Arg, Ala, Gly, Lys, Pro, and Val in the ligand interacting with Asp, Gln, Lys, Ser, Thr, and Leu in the receptor (Appendix A). The decomposition diagram (D) indicated that Arg3 in the peptide enhanced binding, whereas Arg35, Lys931, and Lys1051 generated an adverse intramolecular effect on the total interaction energy.

Molecular docking studies demonstrated that the short peptide (ShortPep) exhibits higher binding affinity and more stable interactions with the key receptors involved in wound healing—VEGFR, EGFR, and TGFRβ2—compared to the long peptide (LongPep). Specifically, the ShortPep displayed binding affinities of −6.7 kcal/mol for VEGFR, −7.2 kcal/mol for EGFR, and −5.6 kcal/mol for TGFRβ2. In all cases, the root mean square deviation (RMSD) values were below 2 Å, indicating stable and reproducible docking poses. These findings suggest that the ShortPep may interact more effectively with growth factor receptors, potentially contributing to its enhanced regenerative activity [53]. Energy decomposition analyses revealed that electrostatic interactions, particularly between arginine residues in the peptide and acidic residues on the receptor, such as aspartate and histidine are critical for stabilizing the EGFR–ShortPep complex. By contrast, the LongPep exhibited unfavorable intramolecular interactions that significantly diminished its overall binding energy. Furthermore, the alluvial diagrams for TGFRβ2 and VEGFR confirmed that the ShortPep establishes more effective hydrogen bonds and hydrophobic contacts, aligning with its superior performance in promoting the cell proliferation and migration observed in in vitro assays. These findings suggest that the structural features and high cationic charge of the ShortPep enhance its ability to interact with the growth factor receptors, thereby supporting its potential to accelerate wound healing, especially in clinical scenarios complicated by infection.

### 2.6. Integration of the In Vitro and In Silico Results

The in vitro and in silico data collectively demonstrate that the short peptide (ShortPep) outperforms the long peptide (LongPep) in terms of cytotoxicity, cell proliferation, and migration, correlating with its higher binding affinity for the key receptors involved in wound healing, namely EGFR, TGFRβ2, and VEGFR. In cytotoxicity assays, the ShortPep exhibited lower toxicity, while in the proliferation and migration assays, it more effectively enhanced mitochondrial activity and promoted wound closure in the Detroit 551 fibroblasts. Consistently, molecular docking studies performed with AutoDock Vina showed that the ShortPep had binding affinities of −6.7, −7.2, and −5.6 kcal/mol toward VEGFR, EGFR, and TGFRβ2, respectively, with the RMSD values below 2 Å, indicating stable and reproducible interactions. Energy decomposition analyses further revealed that electrostatic interactions between the ShortPep’s arginine residues and acidic receptor residues (e.g., aspartate and histidine) are the key contributors to the stability of the EGFR–ShortPep complex. By contrast, the LongPep displayed intramolecular interactions that were energetically unfavorable, significantly reducing its overall binding energy [14,17,18]. Moreover, the alluvial diagrams for TGFRβ2 and VEGFR confirmed that the ShortPep establishes more efficient hydrogen bonds and hydrophobic interactions compared to the LongPep, which is consistent with its superior ability to promote cell migration and proliferation in in vitro assays [40]. These findings suggest that the structural characteristics and cationic charge of the ShortPep enhance its interaction with the growth factor receptors, reinforcing its therapeutic potential in accelerating wound-healing processes [14,17,18]. When combined with its demonstrated antimicrobial activity [32,33], the ShortPep emerges as a promising dual-action candidate for the treatment of chronic ulcers and infections commonly associated with impaired tissue repair.

### 2.7. Clinical Implications

The findings of this study have significant clinical implications for the treatment of chronic wounds, including diabetic foot ulcers, pressure ulcers, and other lower limb lesions. The ability of the short peptide to effectively promote fibroblast proliferation and migration, coupled with its lower cytotoxicity, indicates that it can accelerate re-epithelialization and tissue regeneration. This is critical in clinical scenarios where slow healing increases the risk of infection and complicates wound management [54,55].

Additionally, the robust in silico interaction of the short peptide with critical receptors, such as EGFR, TGFRβ2, and VEGFR, supports its potential to activate signaling pathways that coordinate tissue repair. Activating these receptors fosters not only the proliferation of the key cells in cutaneous regeneration but angiogenesis, which supplies nutrients and oxygen to the healing tissue. Consequently, the therapeutic use of this peptide may improve wound vascularization, leading to more rapid and effective healing [9,54].

Another clinically relevant aspect is the confirmed antimicrobial activity of these peptides. In chronic wounds, bacterial colonization and infection are common complications that significantly delay the healing process. The dual capability of the short peptide to stimulate tissue repair and combat multidrug-resistant pathogens (such as *Pseudomonas aeruginosa* and *Klebsiella pneumoniae*) offers an integrated therapeutic advantage. This could translate into treatments that not only enhance wound closure but prevent or control infections, thereby reducing the reliance on conventional antibiotics and the associated risk of developing resistance [4,56,57,58].

Finally, the stability of the receptor–ligand interactions predicted by molecular docking, attributed to the cationic charge and helical structure of the short peptide, opens avenues for optimizing topical therapeutics. Such formulations could include controlled-release systems to ensure peptide bioavailability and efficacy at the wound site. In summary, combining wound healing and antimicrobial properties makes the short peptide a promising therapeutic tool for managing chronic wounds, though further preclinical and clinical studies are needed to confirm its efficacy and safety in patients.

### 2.8. Limitations and Recommendations

Despite the promise of these cationic peptides in promoting wound healing, this study has certain limitations that must be considered when interpreting the results. First, the in vitro assays with the Detroit 551 fibroblasts did not fully replicate the complexity of the in vivo tissue environment, where multiple cell types, the ECM, and immune factors all play pivotal roles. Moreover, molecular docking studies, while providing valuable insights into binding affinity and atomic-level stability, are computational predictions that require experimental validation in cellular and animal models. Another limitation involves the variability of the concentrations tested; although the thresholds for cytotoxicity and proliferation were identified, these results must be extrapolated to clinical contexts with caution.

Based on these limitations, we recommend expanding the research to include in vivo models of chronic wounds to evaluate the wound-healing efficacy of these peptides. Additionally, further assays should explore how these peptides affect other aspects of healing, such as angiogenesis and inflammatory modulation, and various cell types present in the skin. Validation of receptor–ligand interactions using experimental techniques, like co-immunoprecipitation or atomic force microscopy, would also help confirm in silico findings. Finally, topical formulation and administration of these peptides, as well as potential combinations with other therapeutic agents, should be explored to optimize their regenerative and antimicrobial effects in clinical settings.

## 3. Materials and Methods

### 3.1. Design and Synthesis of the AMPs

Two cationic antimicrobial peptides (CAMPs) were used for sample design and synthesis: a long 39-amino-acid peptide (referred to as ∆M2), which was previously described in references [32,59], and a short 18-amino-acid peptide derived from the ∆M2 Cec D-like analog (CAMP-CecD), which was previously described in reference [33]). To characterize the physicochemical properties of the peptides (ShortPep and LongPep), three online bioinformatics tools were used. PepCalc (https://pepcalc.com/, accessed 15 May 2025) was used to determine the net charge of each peptide at physiological pH, and to obtain general information, such as sequence length and solubility properties. The ExPASy ProtParam tool (https://web.expasy.org/protparam/, accessed 15 May 2025) was used to calculate the aliphatic index and the grand average of hydropathicity (GRAVY). These values provide insight into the peptides’ thermal stability and overall hydrophilic or hydrophobic character. HeliQuest version 2 (http://heliquest.ipmc.cnrs.fr/, accessed 15 May 2025) was used to estimate mean hydrophobicity (H) and to evaluate the peptides’ propensity to form α-helical structures, which is a key factor in many bioactive peptides, including those involved in wound healing. Table 2 presents the physicochemical properties of the peptides (LongPep and ShortPep). Both peptides were purchased from GenScript Corporation (Piscataway, NJ, USA), with a purity level of 98%. Each lyophilized AMP was dissolved in phosphate-buffered saline (pH 7.4; 138 mM NaCl, 3 mM KCl, 1.5 mM NaH_2_PO_4_, and 8.1 mM Na_2_HPO_4_) at an initial concentration of 5000 µg/mL.

Figure 7 illustrates the predicted structural differences between the ShortPep and the LongPep, providing a visual basis for understanding their different molecular profiles. Panel A reveals that the ShortPep adopts a continuous and well-defined α-helix conformation. By contrast, the LongPep exhibits a more complex and extensive three-dimensional structure, characterized by helical segments interrupted by turns and less structured regions, which is consistent with the presence of residues, such as proline, in its sequence, which limit continuous helical propagation.

Panel B uses helical wheel projections to highlight the differences in amphipathicity. The ShortPep shows a clear separation of hydrophobic and polar/charged residues on opposite sides of the helix, visually confirming its strong amphipathic nature, which is ideal for interacting with membranes. By contrast, the LongPep’s helical wheel, representative of a possible helical segment, shows a less organized distribution of residues in terms of amphipathicity. The separation between the hydrophobic and hydrophilic faces is unclear, and including prolines in a theoretical helical representation highlights their disruptive impact.

These visualized structural differences are key. The ShortPep’s defined amphipathic helix suggests a predominantly interfacial mechanism of action, such as direct interaction with cell membranes. By contrast, the LongPep’s more segmented architecture and less pronounced amphipathicity suggest that its comparable biological activity is likely supported by other attributes. These include its larger size, high cationic charge distributed in a complex structure, and capacity for multivalent or diffuse interactions rather than a single dominant amphipathic interface. Thus, the figure shows how different peptide architectures can lead to similar biological functions through different molecular strategies.

### 3.2. Cytotoxicity

The cytotoxicity of the antimicrobial peptides (AMPs) on human skin fibroblasts (Detroit 551, ATCC^®^ CCL-110) was measured using the 3-(4,5-dimethylthiazol-2-yl)-2,5-diphenyltetrazolium bromide (MTT) method. The cells were seeded in 96-well plates and allowed to adhere for 24 h at 37 °C and 5% CO_2_. Then, serial dilutions of each peptide were prepared at a ratio of 1:4 (from 1 µg/mL to 1024 µg/mL) and added to the cells using Dulbecco’s Modified Eagle Medium (DMEM, Sigma, St. Louis, MA, USA) with 10% fetal bovine serum (FBS, Invitrogen, Thermo Fisher Scientific, Waltham, MA, USA) as the culture medium. After 72 h of incubation at 37 °C and 5% CO_2_, the MTT reagent was added [60,61]. Following a 4-h incubation at 37 °C, dimethyl sulfoxide (DMSO, gChem, Covington, LA, USA) was added. The absorbance at 570 nm was then measured using a Varioskan spectrophotometer (Thermo Fisher Scientific, Waltham, MA, USA). Cell viability percentages were calculated based on the obtained optical densities. The assays were performed twice independently; each were conducted in triplicate.

### 3.3. Proliferation

The proliferation of the Detroit 551 cells (ATCC^®^ CCL-110, Manassas, VA, USA) cultured with AMPs was measured using the alamarBlue™ method [48]. The cells were seeded in 96-well plates and allowed to adhere for 24 h at 37 °C and 5% CO_2_. After this period, the cells were treated with the following concentrations of AMPs: 500 µg/mL (more toxic for both peptides), 125 µg/mL (toxic for the long peptide and nontoxic for the short peptide), and 31.2 µg/mL (less toxic for the long peptide and nontoxic for the short peptide). The cells were then incubated for 72 h at 37 °C and 5% CO_2_ with readings taken every 24 h. At each time point, alamarBlue was added (Bio-Rad Laboratories, Hercules, CA, USA). After a 4-h incubation, the fluorescence intensity was measured. Cell proliferation percentages were then calculated relative to the fluorescence intensity in the untreated control cells. The assay was performed using a Thermo Scientific™ Varioskan™ LUX multimode microplate reader (Thermo Fisher Scientific, Waltham, MA, USA). The experiments were performed twice independently, each in triplicate.

### 3.4. Wound-Healing Potential of the Peptides

To evaluate the capacity of the AMPs to induce tissue repair, the Detroit 551 human fibroblasts were cultured in DMEM supplemented with 10% FBS, 100 U/mL penicillin, and 100 µg/mL streptomycin. The in vitro CytoSelect™ 24-Well Wound Healing Assay (Cell Biolabs, Inc., San Diego, CA, USA) was used for this evaluation. The AMPs were tested at concentrations of 4 µg/mL, 16 µg/mL, and 64 µg/mL. DMEM with 10% FBS served as the wound-healing control. The CytoSelect™ system contains a plastic insert in each well that creates a gap in the cell monolayer, thus simulating a skin rupture. In each well with the insert, 250,000 cells/mL were seeded in DMEM supplemented with FBS and antibiotics. The plates were then incubated at 37 °C and 5% CO_2_. After 24 h, the inserts were removed and an initial photograph of the gap was taken. A fresh medium containing the compounds at the aforementioned concentrations was then added, and the cells were incubated again at 37 °C and 5% CO_2_. Photographs were taken at 0 and 24 h, and gap size and percentage of closure were measured using NIS-Elements imaging software 20.0.245 (Nikon, Tokyo, Japan) [62].

### 3.5. In Silico Peptide–Receptor Preparation

#### 3.5.1. Receptor Preparation: TGFRB2

The genetic sequence for TGFRB2 was obtained from GenBank (Accession ID: 7048, consulted on 21 November 2024) to ensure precise modeling of the target protein.

#### 3.5.2. 3D Modeling Using AlphaFold

The sequence was processed in AlphaFold (https://alphafold.ebi.ac.uk/, accessed 25 November 2024), an advanced tool for predicting protein structures. The resulting PDB file was downloaded and visually inspected for structural accuracy.

#### 3.5.3. Validation of the Predicted Structure

ProSA-Web (https://prosa.services.came.sbg.ac.at/prosa.php, accessed 25 November 2024) was used to assess the overall quality of the structure and to identify potential structural anomalies. MolProbity (http://molprobity.biochem.duke.edu/, accessed 25 November 2024) was used to analyze the bond angles, steric clashes, and rotamer misalignments.

### 3.6. Ligand Preparation

#### Construction and Optimization

Ligands were designed using ChemDraw (https://revvitysignals.com/products/research/chemdraw, accessed on 25 November 2024). The structures were optimized with Avogadro version 1.2 (https://avogadro.cc/, accessed on 25 November 2024) using the Universal Force Field, and were subsequently saved in the PDB format to ensure compatibility with the docking tools [4,47,63].

### 3.7. Docking Configuration

AutoDock Vina was used for docking, with a configuration file specifying energy_range = 4 and exhaustiveness = 8. The grid-box dimensions were determined using the CB-Dock tool (http://cao.labshare.cn/cb-dock/, accessed on 25 November 2024). Receptors were prepared in AutoDock Tools (http://autodock.scripps.edu, accessed on 25 November 2024) by adding Kollman charges and optimizing the hydrogen bonds.

#### 3.7.1. Molecular Docking Process

Docking simulations were conducted using AutoDock Vina. Each ligand was docked to the TGFRB2 receptor using the specified parameters. The conformer with the lowest binding energy that met the energy range and the RMSD thresholds was selected.

#### 3.7.2. Docking Validation

A triple redocking procedure was performed to assess the reproducibility of the docking protocol by measuring the mean RMSD values and standard deviations for each of the three runs. As a complementary validation method, cross-docking with a reference ligand (positive controls) previously characterized experimentally was used to verify the consistency and accuracy of the docking protocol.

#### 3.7.3. Post-Docking Analysis

Discovery Studio Visualizer (http://accelrys.com, accessed on 25 November 2024) was employed to visualize the interactions, identifying the key residues involved in the hydrogen bonds, hydrophobic contacts, and electrostatic interactions. Energy decomposition tools highlighted the residues that significantly contributed to the binding energy.

#### 3.7.4. Figures and Results

Alluvial diagrams were built to depict the frequency and type of ligand–receptor interactions, constructed using R version 4.3.1 (https://cran.r-project.org/bin/windows/base/old/4.3.1/, accessed on 25 November 2024).

### 3.8. Statistical Analysis

For the data from the cytotoxicity and proliferation assays in the Detroit 551 human skin fibroblasts, which measured the percentage of cell viability in the presence of short and long AMPs at different concentrations, the average was used as a summary measure, with its respective standard deviation, since the data was normally distributed. Cell migration (wound closure) assays were summarized using medians and interquartile ranges (IQRs) since the data were not normally distributed. These medians were then converted to percentages with their respective 95% confidence intervals to visualize the percentage of wound closure in the different treatments. The *t*-test was used to compare the means and the non-parametric Kruskal–Wallis test to compare the medians for statistical comparisons between the control group and the various treatments (short and long peptides) at different concentrations and times. All analyses were considered statistically significant, with a *p*-value of less than 0.05. Data processing was performed using R 4.4.2 (2024) (open-source software accessed on 25 November 2024) to ensure clear and reliable detection of significant differences in the viability, proliferation, and wound-healing potential of AMPs.

## 4. Conclusions

The findings of this study reveal that the short peptide derived from *G. mellonella* cecropin D exhibits a superior toxicity profile and an enhanced ability to stimulate cell proliferation and migration. This correlates with its markedly higher affinity for the key receptors involved in wound healing (EGFR, TGFBR2, and VEGFR). In vitro assays and in silico molecular docking analyses both demonstrated that the ShortPep establishes stable, robust interactions with favorable energetic contributions compared with the LongPep. These results suggest that the structure and cationic charge of the ShortPep facilitate binding to the growth factor receptors, thereby promoting re-epithelialization and tissue regeneration. Furthermore, given its confirmed antimicrobial activity, the ShortPep emerges as a promising candidate for the treatment of chronic ulcers and infections associated with tissue repair. However, in vivo studies and experimental validation of receptor–ligand interactions are recommended to confirm these findings and to explore potential formulations that optimize its clinical application.

## Figures and Tables

**Figure 1 antibiotics-14-00651-f001:**
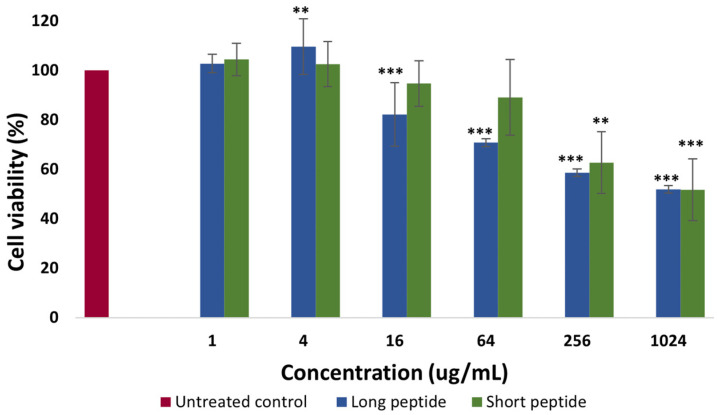
Effect of peptides on the viability percentage of fibroblasts (Detroit 551), with different concentrations of the long peptide and short peptide. With a significance level of *** = *p* value < 0.001; ** = *p* value < 0.01.

**Figure 2 antibiotics-14-00651-f002:**
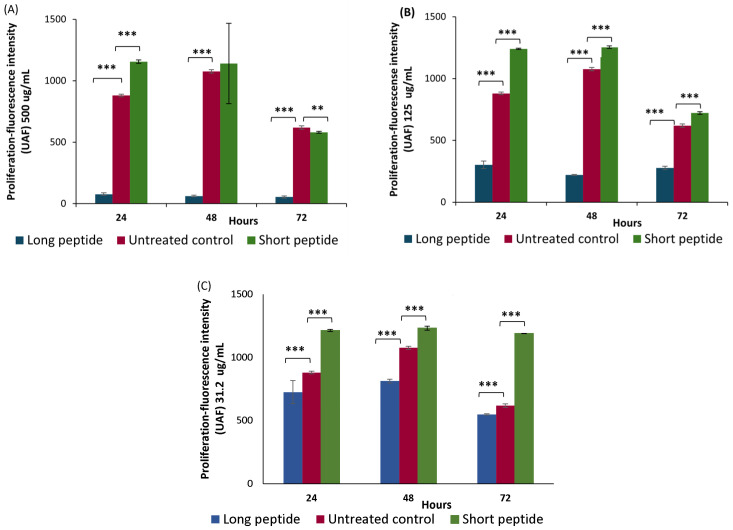
Effect of Long and Short Antimicrobial Peptides on the Proliferation of the Detroit Cells vs. the Untreated Control Cells. The bars show the average fluorescence units obtained in the control cells versus the cells treated with long and short peptides after 24, 48, and 72 h of exposure at doses of (**A**) 500 μg/mL, (**B**) 125 μg/mL, and (**C**) 31.2 μg/mL. Significance levels are as follows: *** = *p* value < 0.001, ** = *p* value < 0.01.

**Figure 3 antibiotics-14-00651-f003:**
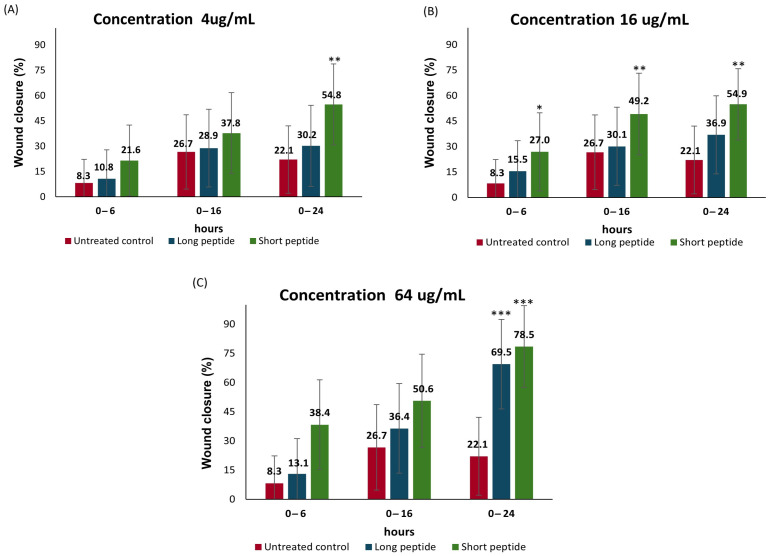
Effect of short and long peptides on wound closure. (**A**) Evaluation of the wound closure capacity of peptides at a concentration of 4 μg/mL. (**B**) Evaluation of the wound closure capacity of peptides at a concentration of 16 μg/mL. (**C**) Evaluation of the wound closure capacity of peptides at a concentration of 64 μg/mL, compared to the control untreated cells. With a significance level *** = *p* value < 0.001; ** = *p* value < 0.01; * = *p* value < 0.05.

**Figure 4 antibiotics-14-00651-f004:**
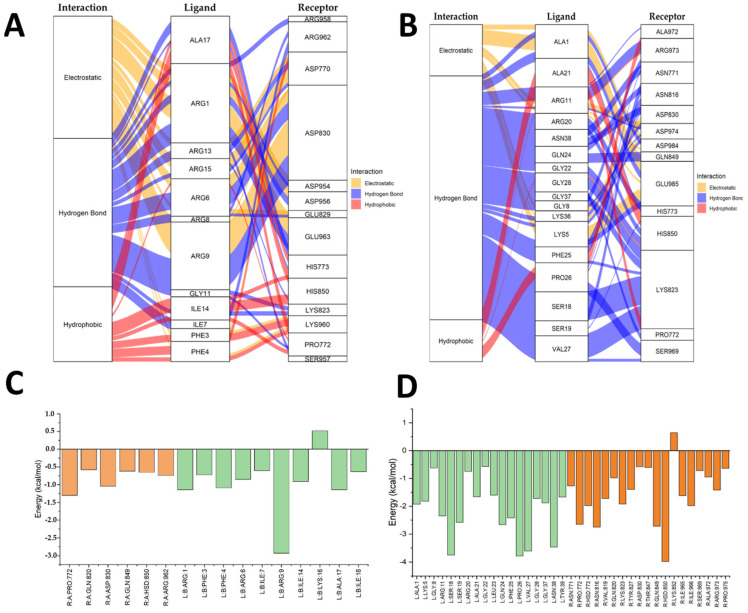
Alluvial diagrams and energy decomposition analysis of residue interactions in the EGFR–ShortPep and the EGFR–LongPep complexes. (**A**) Alluvial diagram of amino acid residue interactions in the EGFR–ShortPep complex. (**B**) Alluvial diagram of amino acid residue interactions in the EGFR–LongPep complex. (**C**) Total per-residue energy decomposition contributions in the EGFR–ShortPep complex. (**D**) Total per-residue energy decomposition contributions in the EGFR–LongPep complex. In the alluvial diagrams, yellow indicates electrostatic interactions, blue represents hydrogen bonds, and red corresponds to hydrophobic interactions. In the bar charts, orange bars denote receptor residues, while green bars indicate ligand (peptide) residues.

**Figure 5 antibiotics-14-00651-f005:**
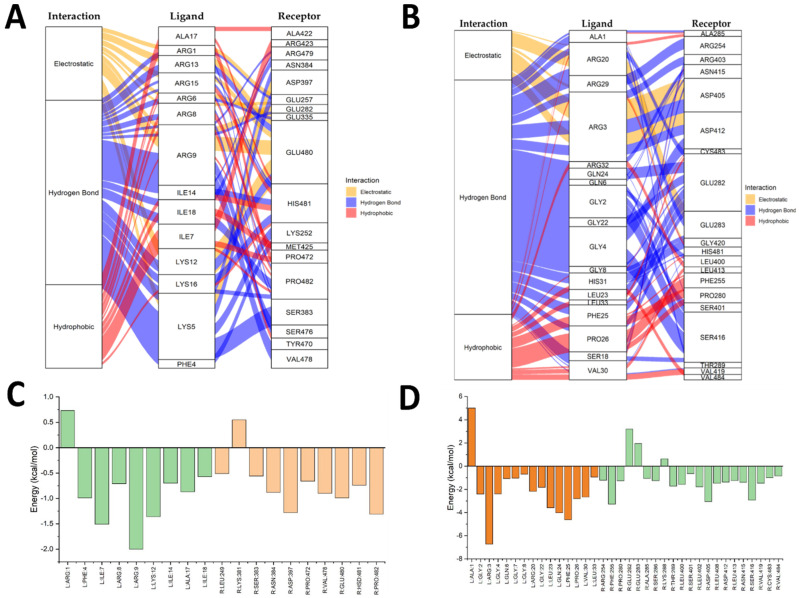
Alluvial diagrams and energy decomposition contributions of the residue interactions in the TGFRB2–ShortPep and the TGFRB2–LongPep complexes. (**A**) Alluvial diagram of the amino acid residue interactions in the TGFRB2–ShortPep complex. (**B**) Alluvial diagram of the amino acid residue interactions in the TGFRB2–LongPep complex. (**C**) Total energy decomposition contributions by the residues in the TGFRB2–ShortPep complex. (**D**) Total energy decomposition contributions by the residues in the TGFRB2–LongPep complex. In the alluvial diagrams, yellow represents electrostatic bonds, blue represents hydrogen bonds, and red represents hydrophobic interactions. In the bar charts 3C, orange denotes the receptor amino acids and green denotes the ligand amino acids. In the bar charts 3D, orange denotes the ligand amino acids and green denotes the receptor amino acids.

**Figure 6 antibiotics-14-00651-f006:**
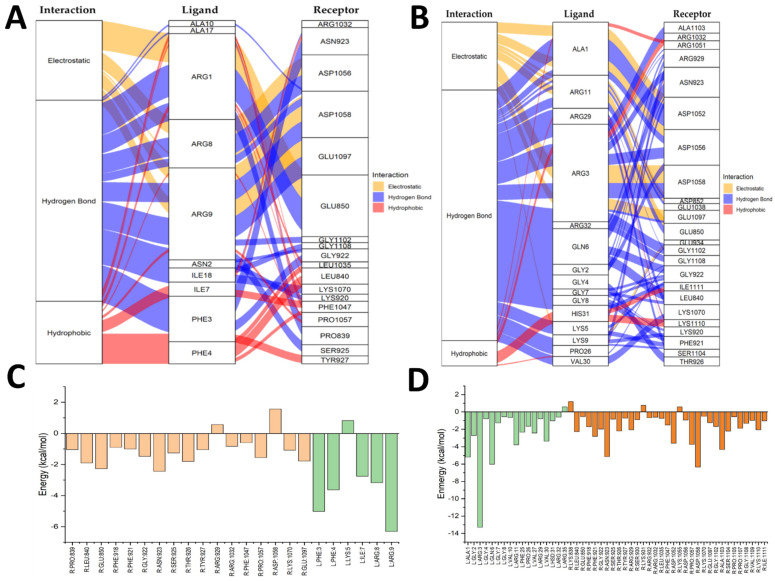
Alluvial diagrams and energy decomposition contributions of the residue interactions in the VEGFR–ShortPep and the VEGFR–LongPep complexes. (**A**) Alluvial diagram showing amino acid residue interactions in the VEGFR–ShortPep complex. (**B**) Alluvial diagram of amino acid residue interactions in the VEGFR–LongPep complex. (**C**) Total energy decomposition contributions by individual residues in the VEGFR–ShortPep complex. (**D**) Total energy decomposition contributions by individual residues in the VEGFR–LongPep complex. In the alluvial diagrams, interaction types are color-coded as follows: yellow for electrostatic bonds, blue for hydrogen bonds, and red for hydrophobic interactions. In the bar charts, orange bars represent receptor (VEGFR) amino acids, while green bars denote ligand (peptide) amino acids.

**Figure 7 antibiotics-14-00651-f007:**
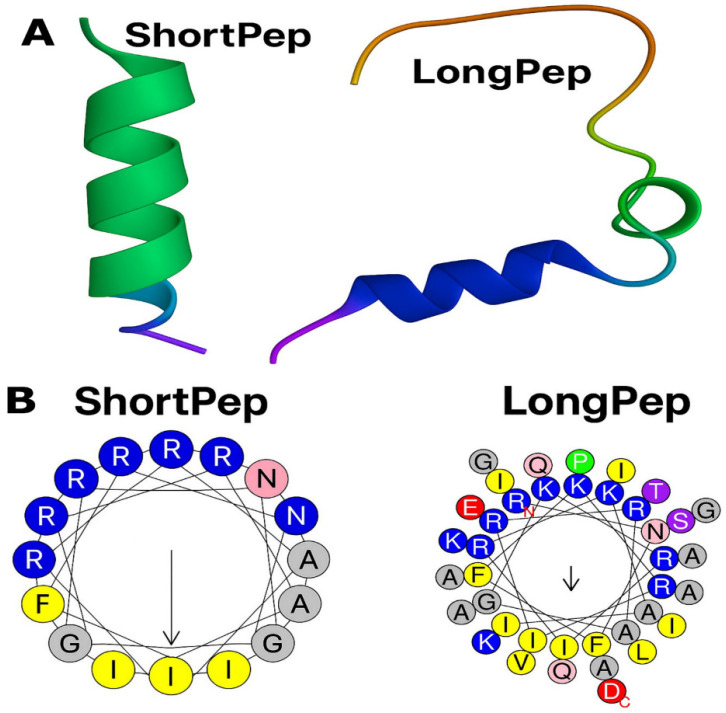
Structural Prediction and Amphipathicity Analysis of ShortPep and LongPep Using Mo-lecular Modeling and Helical Wheel Projections. (**A**) Molecular modeling of ShortPep and LongPep. (**B**) Helical wheel projections of ShortPep and LongPep.

**Table 1 antibiotics-14-00651-t001:** Docking results (AutoDock Vina) showing the mean binding affinity (kcal/mol ± SD) and RMSD (<2 Å).

Receptor	Ligand	Affinity (kcal/mol) ± SD	RMSD (Å)
VEGFR	Axitinib	−7.8 ± 0.3	1.2 ± 0.1
VEGFR	ShortPep	−6.7 ± 0.4	1.6 ± 0.2
VEGFR	LongPep	−5.3 ± 0.2	1.8 ± 0.2
EGFR	Osimertinib	−7.6 ± 0.3	1.2 ± 0.1
EGFR	ShortPep	−7.2 ± 0.5	1.7 ± 0.2
EGFR	LongPep	−3.7 ± 0.4	1.5 ± 0.1
TGFRβ2	Galunisertib	−6.8 ± 0.3	1.1 ± 0.1
TGFRβ2	ShortPep	−5.6 ± 0.2	1.4 ± 0.2
TGFRβ2	LongPep	−4.0 ± 0.3	1.7 ± 0.3

RMSD: root mean squared deviation; SD: standard deviation; ShortPep: short peptide; LongPep: long peptide.

**Table 2 antibiotics-14-00651-t002:** Physicochemical Properties of Short and Long Peptides Used in Wound-Healing Analysis.

Peptide	Sequence	Length	Net Charge	Aliphatic Index	Grand Average of Hydropathicity (GRAVY)	Hydrophobicity <H>	Hydrophobic Moment <µH>
Short Pep	RNFFKRIRRAGKRIRKAI	18	9	76.11	−1.106	−0.002	0.723
Long Pep	RNFFKRIRRAGKRIRKAIISAAPAVETLAQAQKIIKGGD	39	9	95.38	−0.387	0.178	0.296

## Data Availability

Data are contained within the article.

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
