# Peer review of "In Vitro and In Silico Wound-Healing Activity of Two Cationic Peptides Derived from Cecropin D in Galleria mellonella"

_antibiotics, 2025, doi:10.3390/antibiotics14070651_

Round 1
Reviewer 1 Report
Comments and Suggestions for Authors
In the manuscript, S. Rivera-Sanchez et al. employed both experimental and computational methods to investigate the wound-healing activity of two cationic peptides derived from Cecropin D of Galleria mellonella. The authors assessed the cytotoxicity, proliferation, and migration effects of the two peptides, complemented by molecular docking studies on three key receptors associated with wound healing. This research holds potential clinical implications for the treatment of chronic wounds. I recommend considering this work for publication once the following major concerns are adequately addressed.
- In proliferation assays, the authors indicate that the experiments were conducted twice independently, each with triplicate measurements. Consequently, for each experimental condition, there should be a total of 6 measured values. However, in Figure 1, only the median is presented in the bar plot along with the calculated p-value. To enhance the clarity of the data, I recommend including error bars in the bar plot to provide the audience with insight into the distribution of the data.
- In the cell migration assay, the author mentioned that “By 16 h, further gap reduction was observed, most notably at 64 μg/mL, where the short peptide achieved a 55% closure compared to 36% for the long peptide, with statistically significant differences between the two (p = 0.049).” Given that a p-value of 0.049 is very close to the conventional significance threshold of 0.05, it may not be appropriate to assert significant differences between the short and long peptides. Similarly, Table 2 shows a p-value of 0.049 in the context of multiple comparisons. I suggest the authors carefully consider the interpretation of these results
- The x-tick labels in Figures 2, 3, and 4 are too small, making it difficult to discern the specific residues and their indices of the ligand and receptor. I recommend increasing the font size of these labels for better readability.
- In Figure 2, the legend specifies that yellow represents electrostatic interactions, blue represents hydrogen bonds, and red denotes hydrophobic interactions. However, there is a discrepancy with the caption and text (lines 304-305), which state, “Blue indicates electrostatic interactions, red denotes hydrogen bonds, and yellow represents hydrophobic interactions.” This inconsistency is also present in Figures 3 and 4.
- In the docking study for peptide-TGFRB2, the authors state, “Ala1, Gln280, and Gln281 ultimately decreased affinity, indicating that their intramolecular participation lowers the complex’s global stability” (lines 334-335). But in Figure 3D, there is no Gln 280 and Gln 281.
- The author conducted docking study for both peptides across three different receptors and calculated the binding affinity, I suggest showing the binding poses for both peptides on the three receptors. Additionally, identifying key residues within the binding pockets of the receptors would enhance the interpretation of the results.
- For the molecular docking results, the authors performed the docking procedure three times and reported the mean and standard deviation of the binding energy. However, for the root mean square deviation (RMSD), only a single value is provided. I recommend including the mean and standard deviation for RMSD to provide a more comprehensive overview of the results.
- In the introduction section, I recommend that the authors offer a summary of recent studies on wound-healing activities and the existing challenges faced in this area. This context would help frame the significance of their work.
Author Response
Dear Reviewer
Please check the attached document

Reviewer 2 Report
Comments and Suggestions for Authors
Using a cell model and molecular docking, the authors developed and tested two peptides (one long and one short peptide) on their effects on wound healing activity. The authors should address the following comments before considering acceptance:
- in the molecular docking session, the authors named peptides ShortPep and LongPep, but not at the start of the manuscript; make sure to have the unique names all across the manuscript and results;
- The sequence, MW, and purity of two peptides are missing.
- The results contain many incorrect statistical analyses. In Figure 1, the most common p-value seems to be 0.02, even with a small and huge difference across different groups. That is not correct. The author should make sure all data is analyzed correctly. Table 1's statistical analysis is missing.
- In Table 2, at 0 hours, there is already a difference, which means the starting cells are not the same. Since the starting points are different, all the following results are not correct. The authors should re-conduct this part to ensure the data is scientifically robust.
- All assays are conducted under ug/mL, and the long peptide is twice as big as the short peptide, meaning the concentration is twice as different, and the results cannot be directly compared.
Author Response

(The authors gave the same response as above.)

Reviewer 3 Report
Comments and Suggestions for Authors
The manuscript presents an excellent and timely study, wherein the author has screened peptides derived from Cecropin D for their wound healing properties. The experimental approach is relevant, and the findings contribute meaningfully to the growing field of peptide-based therapeutics. Below are specific comments and suggestions to improve the clarity and scientific rigor of the manuscript
ï‚· Keywords: Please arrange the keywords in alphabetical order, as per journal guidelines.
ï‚· Selection of ΔM2 Variant: The rationale for selecting the ΔM2 variant of Cecropin D as the focus of the study requires further clarification. While it is established that other ΔM variants also exhibit antimicrobial activity (J. Oñate-Garzón et al., Journal of Antibiotics, 2017), the manuscript should justify why ΔM2 was prioritized over others.
ï‚· Line 90–92: Please explicitly state that ΔM2 is derived from Cecropin D to aid reader understanding and maintain clarity in the narrative.
ï‚· Cytotoxicity vs. Proliferation (Short Peptide): The results regarding mitochondrial-related activities such as cytotoxicity and proliferation need more in-depth explanation. The short peptide displays cytotoxicity at 256 µg/mL (Table 1), yet proliferation is observed at 500 µg/mL (Figure 1). Similarly, Long peptide shows toxicity at 64 µg/ml but shows proliferation at 125 µg/ml. This apparent contradiction should be addressed mechanistically or experimentally.
ï‚· Proliferation Inconsistencies at 72 Hours: At the 72-hour mark, the short peptide shows proliferation at both 31.2 and 500 µg/mL, but not at 125 µg/mL, where a reduction is seen. Please comment on this non-linear dose response and provide a possible explanation.
ï‚· Dose-Response Curve (Short Peptide, 48 Hours): In Figure 1, the short peptide shows comparable proliferation activity across a broad concentration range (31.2–500 µg/mL) at 48 hours. This suggests a saturation effect or minimal dose-dependency. Please elaborate on how peptide concentration is influencing biological activity in this case.
ï‚· Wound Healing vs. Cytotoxicity (Long Peptide): According to Table 1, the long peptide is cytotoxic at 64 µg/mL. However, it demonstrates a significant 68% wound healing effect at the same concentration within 24 hours. This discrepancy should be clarified, perhaps considering differences in assay sensitivity, cell types, or compensatory mechanisms.
ï‚· Comparison at 16 µg/mL (Both Peptides): Both peptides exhibit comparable wound healing activity at 16 µg/mL after 24 hours. Please discuss this observation in the context of their physicochemical properties, such as amino acid composition, hydrophobicity, and secondary structure.
Author Response

(The authors gave the same response as above.)

Reviewer 4 Report
Comments and Suggestions for Authors
In this manuscript, the authors evaluated the wound healing activity of a 39-mer longer peptide (ΔM2) and an 18-mer shorter peptide (CAMP- CecD). According to their report, the shorter peptide exhibited lower toxicity and enhanced wound healing compared to the longer peptide. Subsequent docking studies with VEGF, TGFR2, and EGFR suggested that the shorter peptide binds with higher affinity. Based on these findings, the authors concluded that the 18-mer peptide is their lead compound for wound healing.
However, the manuscript lacks sufficient supporting data to substantiate these claims.
- The cytotoxicity data presented in Table 1 is not valid for the shorter peptide. The data points are widely scattered, making it difficult to confirm the peptide is not cytotoxic. This data should have been presented as a dose–response or bar chart with appropriate error bars to allow for clear comparison.
- There is no direct evidence supporting the involvement of the VEGF/TGFR2/EGFR axis in wound healing following peptide treatment. Docking studies alone are insufficient. The authors should have included biochemical data demonstrating that the peptides directly interact with VEGF, TGFR2, or EGFR.
- Additionally, proper control data is missing in most experiments. For instance, in the wound healing assay, there is no data representing control group (buffer only), although the authors mention its use in the supplementary information.
- A scrambled version of the peptide could have been used as a negative control to demonstrate that the observed activity is sequence-specific for CAMP- CecD.
- Wound healing images would have been provided, rather than in a tabular format.
Overall, the manuscript suffers from several issues, including poor data representation, lack of appropriate controls, and insufficient data to support the conclusions.
Therefore, I do not consider this manuscript suitable for publication in its current form.
Comments on the Quality of English LanguageThe language could have been more precise.
Author Response

(The authors gave the same response as above.)

Round 2
Reviewer 1 Report
Comments and Suggestions for Authors
The authors did a great job of addressing my comments. Thank you!
Reviewer 2 Report
Comments and Suggestions for Authors
The author has addressed the comments effectively. No further comments.
Reviewer 4 Report
Comments and Suggestions for Authors
Thank you for making the changes. I accept and would recommend the manuscript for publication.